# Biomechanical Comparison between Inverted Triangle and Vertical Configurations of Three Kirschner Wires for Femoral Neck Fracture Fixation in Dogs: A Cadaveric Study

**DOI:** 10.3390/vetsci10040285

**Published:** 2023-04-10

**Authors:** Seonghyeon Heo, Haebeom Lee, Yoonho Roh, Jaemin Jeong

**Affiliations:** 1College of Veterinary Medicine, Chungnam National University, Daejeon 34134, Republic of Korea; 2Institute of Animal Medicine, College of Veterinary Medicine, Gyeongsang National University, Jinju 52828, Republic of Korea

**Keywords:** femoral neck fracture, internal fixation, K-wire configuration, inverted triangle, dog

## Abstract

**Simple Summary:**

Femoral neck fracture in dogs is one of the common orthopaedic diseases, which is usually caused by trauma. As one of the therapeutic options for this, it can be managed through the insertion of metallic implants in the neck. However, a comparative study on the stiffness according to the fashion of the implant insertion into the femoral neck fracture has not been reported in veterinary medicine. Therefore, we compared the mechanical results regarding the insertion of Kirschner wires, pin-type implants, into the femoral neck fracture cadaver model in vertical and inverted triangle fashions. As a result of the experiment, fixation of the fracture site was more stable when K-wires were inserted in an inverted triangle fashion. Additional clinical and in vivo studies are warranted to investigate the efficacy of the inverted triangle configuration of K-wires for femoral neck fracture fixation.

**Abstract:**

The purpose of this study was to compare single-cycle axial load and stiffness between inverted triangle and vertical configurations of three Kirschner wires (K-wires) for femoral neck fracture fixation in small dog cadaveric models. In each of the eight cadavers, the basilar femoral neck fracture model was prepared on both sides of the femur. One side of the femur was stabilized with three 1.0 mm K-wires of an inverted triangle configuration (group T), and the other femur was stabilized with a vertical configuration (group V). Postoperatively, the placement of the K-wires was evaluated with radiographic and computed tomography (CT) images, and static vertical compressive loading tests were performed. The mean yield load and the lateral spread were significantly higher in group T compared to group V (*p* = 0.023 and <0.001). On the cross-section of femoral neck at the level of the fracture line, the surface area between K-wires was significantly larger (*p* < 0.001) and the mean number of cortical supports was significantly higher in group T (*p* = 0.007). In this experimental comparison, the inverted triangle configuration of three K-wires was more resistant to failure under axial loading than the vertical configuration for canine femoral neck fracture fixation.

## 1. Introduction

Often caused by trauma, fractures of the proximal end of the femur including basilar neck and capital physis are common orthopaedic diseases, especially in immature dogs and cats [1,2]. Most affected patients present with non-weight bearing lameness and show pain and crepitation on physical evaluations. These fractures may accompany comminuted proximal femoral fractures, but the majority are simple basilar fractures [2,3]. Conservative treatment of femoral neck fractures commonly results in non-union or hypertrophic pseudoarthrosis with subsequently decreased range of motion and persistent lameness of pelvic limbs. Therefore, surgical intervention is recommended for optimal outcomes [1,2,4,5]. Internal fixation is considered the gold standard for simple femoral neck fractures, and the prognosis ranges from good to excellent following proper fixation [2,4].

The two most common fixation methods for simple femoral neck fractures are a lag screw with anti-rotational pins and three Kirschner wires (K-wires) [1,2,4,6,7,8]. Lag screws have some biomechanical advantages due to their interfragmentary compression forces, resulting in greater stability than three K-wires [2,6,8]. However, theoretically, the interfragmentary compression can induce physis iatrogenic closure, and the use of three K-wires rather than lag screws may be recommended for growing patients [4,9]. The failure rate of fixation of femoral neck fractures has been reported as 11% in dogs, requiring subsequent salvage procedures, such as femoral head and neck ostectomy or total hip replacement [1,10]. Because of the significant shear and bending stress across the fracture plane, successful repair requires proper reduction and selection of stable implants [2,8].

In human medicine, the most common treatment modality for simple femoral neck fracture fixation is the use of three cannulated screws in an inverted triangle configuration [11,12,13]. Several studies have demonstrated that the inverted triangle configuration of three screws is superior to other configurations, resulting in improved stability and low complication rates [14,15,16,17]. Factors contributing to the stability of three cannulated screw fixation in human femoral neck fractures include parallelism, spread, and cortical support [4,12,18]. Although biomechanical studies have demonstrated the mechanical strength of three K-wires in a vertically-oriented configuration for the fixation of femoral neck fractures in dogs, an attempt has yet to be made to compare the different configurations of the three K-wires, resulting in a lack of standard guidance for practitioners [3,6].

The objective of this study was to compare the single-cycle axial load and stiffness between inverted triangle and vertical configurations of three K-wires for femoral neck fracture fixation in dogs. Our hypothesis was that the biomechanical properties of the inverted triangle configuration of three K-wires would be greater than those of the vertical configuration of three K-wires.

## 2. Materials and Methods

### 2.1. Specimens

This cadaveric study was approved by the institutional animal care and use committee of the Chungnam National University (no. 202103A-CNU-038). Eight canine cadavers weighing <10 kg from various breeds euthanatized for reasons unrelated to this study were included in this study. All dogs underwent a thorough orthopaedic examination of the pelvic limbs to ensure the absence of pre-existing orthopaedic diseases. Preoperative computed tomography (CT) scans (Alexion, Toshiba Medical System, Nasu, Japan) were performed to ensure that all dogs had no bone abnormalities in the pelvic limbs. One side of each femur was randomly assigned to each of the two groups in either an inverted triangle (group T) or a vertical (group V) configuration using a coin flip. The cadavers were frozen at −20 °C and thawed to room temperature for 24 h before fixation model preparation [19].

### 2.2. D-Printed Pinning Guides

The preoperative CT images were exported as digital imaging and communication in medicine format and reconstructed using a computer software (3D Slicer, free open-source software platform, http://www.slicer.org, accessed on 13 September 2021) to create three-dimensional (3D) models of the femurs as STL files. Thresholding was performed to identify the cortical bone of the femoral neck (Hounsfield units > 600). The reconstructed 3D models of the femurs were transferred to a computer software (3DS Max, Autodesk, CA, USA), and 3D pinning guides of each femur were designed (Figure 1). Depending on the assignment, three parallel 1.0 mm K-wire holes were created in either an inverted triangle or a vertical configuration.

For a 3D pinning guide of group T, the first inferior K-wire hole was created caudal and distal to the greater trochanter at the level of the third trochanter. Then, two superior parallel K-wire holes were created, constructing an inverted equilateral triangle. For a 3D pinning guide of the opposite side of the femur, group V, the first inferior K-wire hole was created in the same manner as for group T, and two superior parallel K-wire holes were created equidistant apart, constructing three vertically oriented K-wire holes. The position of each K-wire was set so that the distance between each wire spread the widest in the range that did not invade the femoral neck cortex in the narrowest section of the femoral neck.

To prevent dislocation of the 3D pinning guide, two side arms and a temporary 0.8 mm K-wire hole were made in the proximal and distal portions, respectively. The pinning guides were printed using a 3D printer (Finebot-Z420, TPC Mecatronics, Seoul, Korea) with polylactic acid filament.

### 2.3. Preparation of Femoral Neck Fracture Model

Eight pairs of femurs were obtained and carefully stripped of all remaining soft tissues. The basilar femoral neck fracture model was created for each femur using an oscillating saw from the medial aspect of the greater trochanter proximal-lateral to the lesser trochanter distal-medially. The fragment was held in an anatomical position with pointed, bone-holding forceps, and the femoral neck fracture angles were measured according to a previously described Pauwels angle using a computer software (vPOP-PRO version 2.3.9, VetSOS Education Ltd., Shrewsbury, UK) [20]. In brief, the anatomic axis of the femoral shaft was determined in the anteroposterior view of the femur. Then, a line perpendicular to the anatomic axis was drawn at the level of the femoral head. The angle between this line and the neck fracture line was defined as a neck fracture angle.

### 2.4. Fixation Model Preparation

A 3D printed pinning guide was secured with two temporary 0.8 mm K-wires; then, three 1.0 mm K-wires for fixation were inserted into each femur until resistance of the far cortex of the femoral head was felt. The three 1.0 mm K-wires for fixation were cut leaving 3 cm, and the 0.8 mm temporary K-wires and a 3D printed pinning guide were removed. One surgeon (J.J.) performed all surgical procedures. The specimens were wrapped individually in 0.9% saline-soaked cotton gauze and frozen at −20 °C until mechanical testing.

### 2.5. Spread of the K-Wires

Postoperative anteroposterior and lateral radiographs of the femur were obtained. When obtaining lateral projection radiographs, the femurs were tilted to an abduction angle of 45° so that the femoral neck region was visible [21]. The spread of K-wires in both views was measured using a previously described method [22]. Briefly, the percentage of the distance between the borders of the outer K-wires to the width of the femoral neck at the fracture line level was calculated (Figure 2).

### 2.6. Surface Area of Support Base of K-Wires

Postoperative CT scans were performed for each femur. The CT images were reconstructed, and transverse cross-sections of the femoral neck were obtained. On the cross-section of the femoral neck at the fracture line level, the centres of three K-wires were marked, and the surface area created by the three centres was measured (Figure 3A).

### 2.7. Distance from K-Wire to Femoral Neck Cortex

On the cross-section of the femoral neck at the fracture line level, each K-wire was defined as proximo-caudal, proximal-cranial, and distal in group T, and proximal, middle, and distal in group V. The shortest distance from a K-wire to the adjacent femoral neck cortex was measured (Figure 3B). It was defined as cortical support when the distance from K-wire to the femoral neck cortex was within 1 mm, and the number of K-wires with cortical support was counted for each femur.

### 2.8. Mechanical Testing

A test was designed to estimate the maximum vertical force at the fracture line before failure by simulating the ground reaction force on a hind limb. To ensure the consistency of all specimens, we designed an individual jig capable of an abduction angle of 20°, similar to the normal hip joint angle during stance phase in dogs [6,23]. The distal quarter of each femur was potted to a cylinder container filled with melted resin, and the diaphysis of the femur was held with a 3D-printed jig until the resin fully hardened. The container with a specimen was then secured to the base of a mechanical compressive testing machine (E1000, Instron. Corp., Canton, MA, USA) (Figure 4). An axial compressive load to failure was applied to the femoral head at a constant rate of 50 mm/min through a metallic rod mounted on a load cell that was attached to the crosshead. The test was stopped when placed K-wires were bent and the fracture line was open. Load and crosshead displacement were recorded for each sample. Load at failure was defined as the point at which the first sudden decrease in load occurred on the load-displacement curve. The yield load and displacement were determined by 0.2% offset yield. Stiffness was calculated from the linear portion of the elastic region of the response curve by yield load and displacement. All procedures were recorded, and one investigator (S.H.) determined the modes of failure.

### 2.9. Statistical Analysis

An a priori power analysis was performed using a statistical software (G*Power V3.1.9.2x, Christian-Albrechts-Universität Kiel, Dusseldorf, Germany) to determine the minimum number of dogs required for this study. The minimum sample size of seven pairs of femurs was determined based on: alpha = 0.05, power = 0.95, and an estimated effect size (d = 1.516921) using the mean and standard deviation yield point of femoral neck fracture fixation with inverted triangle and vertical three K-wires in a pilot study of three pairs of femurs. The final sample size was eight pairs of femurs, with anticipation of 10% expected dropout. The statistical analyses of results were performed using the SPSS software version 26.0 (IBM SPSS statistics 26.0, IBM Corp., Chicago, IL, USA). Normal distributions of data were evaluated using the Shapiro–Wilk test. Continuous data (i.e., femoral neck fracture angle, yield point, stiffness, displacement, and values of K-wire placement evaluations) between the two groups were compared using a paired Student’s *t* test. Statistical significance was set at *p* = 0.05. Categorical data (i.e., modes of failure) between the two groups were compared using the McNemar test.

## 3. Results

### 3.1. Descriptive Data

Data were collected from eight pairs of femurs of all dogs. The dogs consisted of one Pomeranian, one Maltese, one Cocker spaniel, two Miniature poodles, and three mixed breeds. The mean body weight of the dogs was 4.8 (range 3.6–6.3) kg. The mean ± standard deviation (SD) femoral neck fracture angles were 64.84 ± 8.67 degrees and 63.93 ± 3.98 degrees in group T and group V, respectively. There was no significant difference in femoral neck fracture angles between the two groups.

### 3.2. Mechanical Testing

The results of mechanical testing are presented as mean ± SD in Table 1. The mean yield load was significantly higher in group T than in group V (*p* = 0.023). Stiffness and displacement showed no significant differences between the two groups.

### 3.3. Postoperative K-Wire Placement Evaluation

The spread and surface area of K-wire placement are presented as mean ± SD in Table 2. The anteroposterior spread of K-wires was not significantly different between the two groups. The lateral spread was significantly high in group T (*p* < 0.001). The surface area was significantly large in group T (*p* < 0.001). The distances from K-wire to cortex are presented as mean ± SD in Table 3. The mean number of cortical supports was significantly higher with 2.75 in group T than 1.75 in group V (*p* = 0.007).

### 3.4. Modes of Failure

All the fixation failed with ventral displacement of the proximal fragment with the fracture line open dorsally. Ventral rotation of the proximal segment was accompanied in three cases (3/8, 37.5%) in group T and four cases (4/8, 50%) in group V. There was no statistically significant difference of failure mode between the two groups (*p* = 1.000).

## 4. Discussion

In this study, the single-cycle axial load between inverted triangle and vertical configurations of three K-wires for femoral neck fracture fixation were compared in cadaveric canine models. The yield load was significantly higher in the inverted triangle configuration than in the vertical configuration, which is consistent with previous studies on humans [16,24]. Therefore, our hypothesis that the biomechanical properties of the inverted triangle configuration of three K-wires would be greater than that of the vertical configuration of three K-wires was accepted.

When a vertical compressive loading is applied to the proximal fragment of the femoral neck fracture, a pivot point appears on the inferior aspect of the femoral neck, the calcar region, around which the proximal fragment is displaced and rotated [25]. The location of a K-wire relative to the pivot point influences fixation stability, and the greater the distance of the K-wire from the pivot point, the longer the lever arm, resulting in greater stability [25]. For three K-wires for femoral neck fracture fixation, the anteroposterior and lateral spread of K-wires can represent the lever arm of the fixation. Gurusamy et al. reported that a reduced spread of three cannulated screws on the lateral view was associated with a significantly increased risk of non-union of the human femoral neck fractures [22]. In our study, the inverted triangle configuration of three K-wires resulted in a significantly higher lateral spread than that of the vertical configuration, suggesting a low potential for mechanical failure of femoral neck fracture fixation in clinical settings.

Several studies have demonstrated that widely spread pins or screws are mechanically more stable than those placed closely in femoral neck fracture fixation [18,22,26]. A greater spread results in a larger surface area of support base of fixation, which in turn, results in larger load distribution and higher area and polar moment of inertia of the fixation [26,27]. The measured mean surface area of the support base was significantly smaller than that of the inverted triangle in our study. Consequently, the single-cycle axial load test showed a higher yield load in the inverted triangle group than in the vertical configuration group.

It is recommended to place the screws close to the cortex of the femoral neck within 3 mm to achieve cortical support of the screws in human femoral neck fracture fixation [11,12,28]. In a previous study, posterior cortical support for the proximal screw of two screws for fixation of human femoral neck fractures was shown to significantly increase loads at 2 mm deflection of the osteotomy site [29]. Later studies also demonstrated a significantly higher union rate in femoral neck fractures with cortical supports of both screws than with the support of one screw or with no cortical support [30,31]. In the proximal region of the femur, two distinct points of bone purchase are the lateral cortex of the femoral neck and the subchondral bone of the femoral head. Positioning of a screw close to the femoral neck cortex provides an additional bone purchase, creating a three-point fixation which increases the stability of fixation [2,31]. For the modification of cortical support of cannulated screws in human medicine, we defined cortical support of K-wires as within 1 mm, considering the relative ratio of the mean femoral neck width in our study to that reported in humans [32,33]. Consequently, the inverted triangle configuration allowed the mean number of cortical supports of K-wire to be 2.75, whereas the vertical configuration was 1.75.

The trabeculae are one of the critical components of proximal femur which determines hip fracture resistance, and the thickness of trabeculae with implant anchorage is a main factor influencing the initial yield behaviours [34,35]. There are two typical trabecular groups, including tensile and compressive in the proximal femur, and the triangular area without trabecular bone at the base of the femoral neck is defined as Ward’s triangle [34]. For the femoral neck fracture fixation, the inverted triangle configuration has the mechanical advantage that only the most distal K-wire passes through the Ward’s triangle as compared with the vertical configuration, where two distal K-wires have a high risk of passing through the Ward’s triangle [16]. Although the distribution of trabeculae of the femoral neck was not demonstrated in our study, our mechanical results are consistent with the previous study and may reflect the effect of trabeculae on stability. Additional studies are needed to investigate the effect of trabecular bone on femoral neck fracture fixation in dogs.

The peak vertical force acting on a pelvic limb during walking is approximately three to four times the body weight in dogs, which can be considered as the minimum reference value for the stability of femoral neck fracture fixation [3,6,36]. In our study, the mean yield point of group T reached four times the mean body weight of the dogs, and these results may indicate that the inverted triangle configuration of three K-wires has sufficient stability for the femoral neck fracture fixation in dogs. Notably, the mean yield load of the vertical configuration of three K-wires did not reach three times the mean body weight of the dogs, contrary to the two previous mechanical studies on vertical three K-wires [3,6]. Possible reasons for the discrepancy between the results of our study and those of previous studies include regular fracture surface created by an oscillating saw and the high femoral neck fracture angle in our study, which is disadvantageous to stability [20,37]. Additionally, the smaller size of dogs and K-wires used in our study may have affected the mechanical results.

This study has some limitations. The ex vivo cadaveric study does not fully represent clinical situations. Postoperative fracture healing and associated complications were not assessed. In addition to mechanical stability, other factors affecting fracture healing, such as vascular preservation for both configurations, should be investigated in further clinical and in vivo studies [4]. Other major limitations of our study were that only vertical axial load was applied to the mechanical testing. Although vertical forces are the main external forces of the canine hip joint during walking, other external and internal forces were not considered [36,38]. Furthermore, the static loading test may not as accurately represent the physiological loading conditions as the cyclic loading test; nonetheless, the static loading test may mimic the condition of early fixation failure due to overload.

## 5. Conclusions

In conclusion, we found that the inverted triangle configuration of three K-wires was more resistant to yield under axial loading than the vertical configuration in canine femoral neck fracture fixation. The inverted triangle configuration had the advantages of greater lateral spread and more cortical supports of three K-wires than those with the vertical configuration. Our results suggest that the inverted triangle configuration of three K-wires is more stable for fixation of canine femoral neck fracture than is vertical configuration. Further investigations into clinical and in vivo studies focusing on the efficacy of the inverted triangle configuration of three K-wires for femoral neck fracture fixation are needed.

## Figures and Tables

**Figure 1 vetsci-10-00285-f001:**
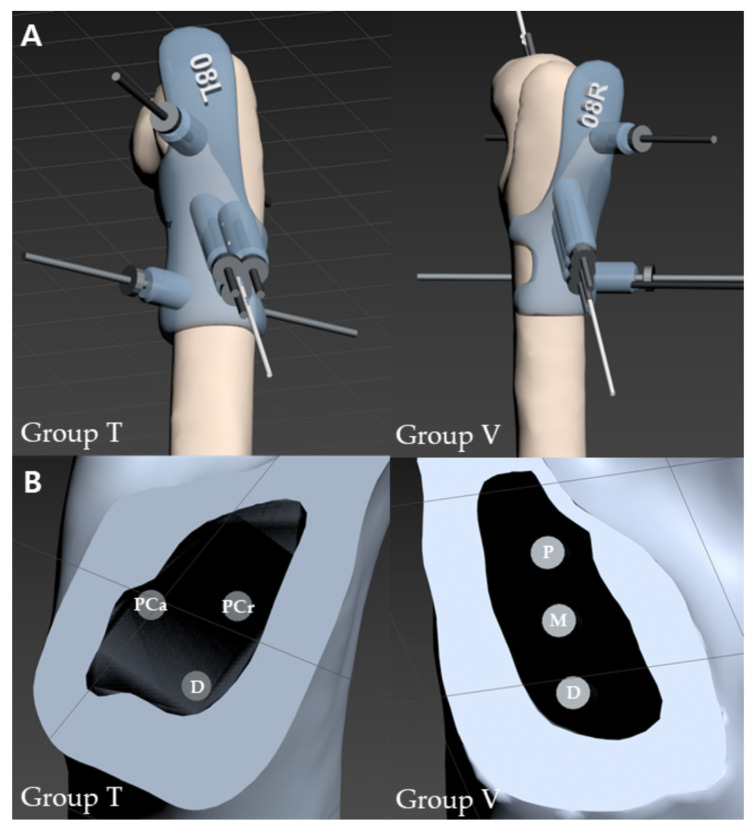
Three-dimensional reconstructed images of pinning guides for each group. (**A**) Three-dimensional pinning guides secured to the lateral surface of the proximal femurs. (**B**) Cross-sections of the femoral neck with K-wire holes. Abbreviations: PCa, proximo-caudal; PCr, proximo-cranial; P, proximal; M, middle; D, distal; T, inverted triangle; V, vertical.

**Figure 2 vetsci-10-00285-f002:**
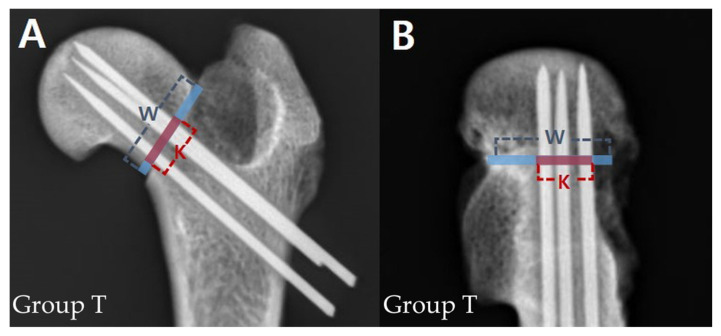
Measurement of anteroposterior (**A**) and lateral (**B**) spread of K-wires in inverted triangle group (Group T). The blue line represents width of the femoral neck at the level of the fracture line (W) and the red line represents distance between the borders of the outer K-wires (K). The spread was calculated as K/W × 100.

**Figure 3 vetsci-10-00285-f003:**
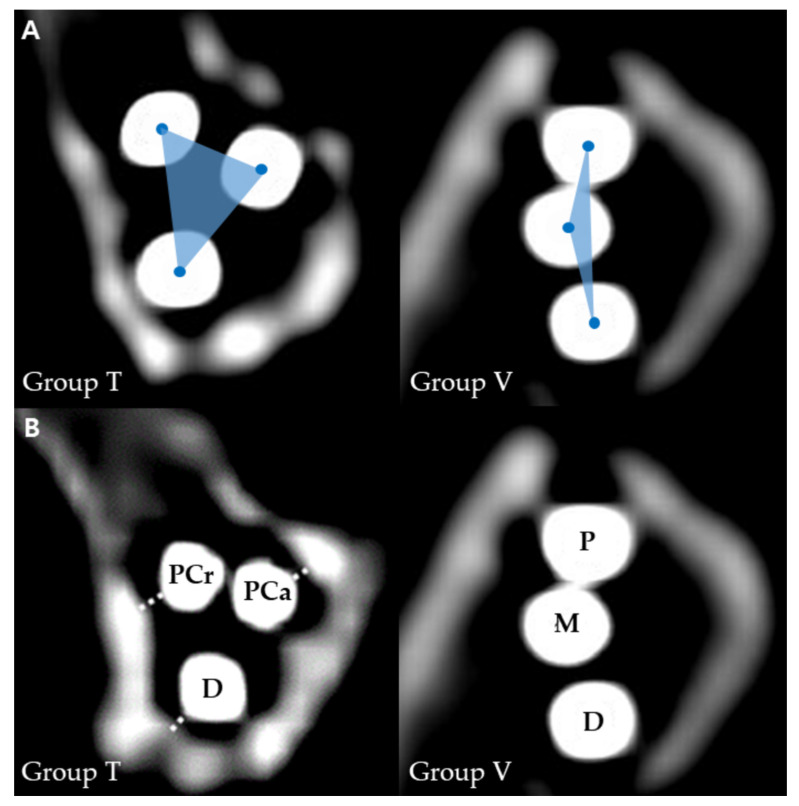
Cross-sections of the femoral neck at the level of the fracture line of each group. (**A**) Surface area of the support base of K-wires. (**B**) The shortest distance from K-wire to the cortex. Abbreviations: PCa, proximo-caudal; PCr, proximo-cranial; P, proximal; M, middle; D, distal; T, inverted triangle; V, vertical.

**Figure 4 vetsci-10-00285-f004:**
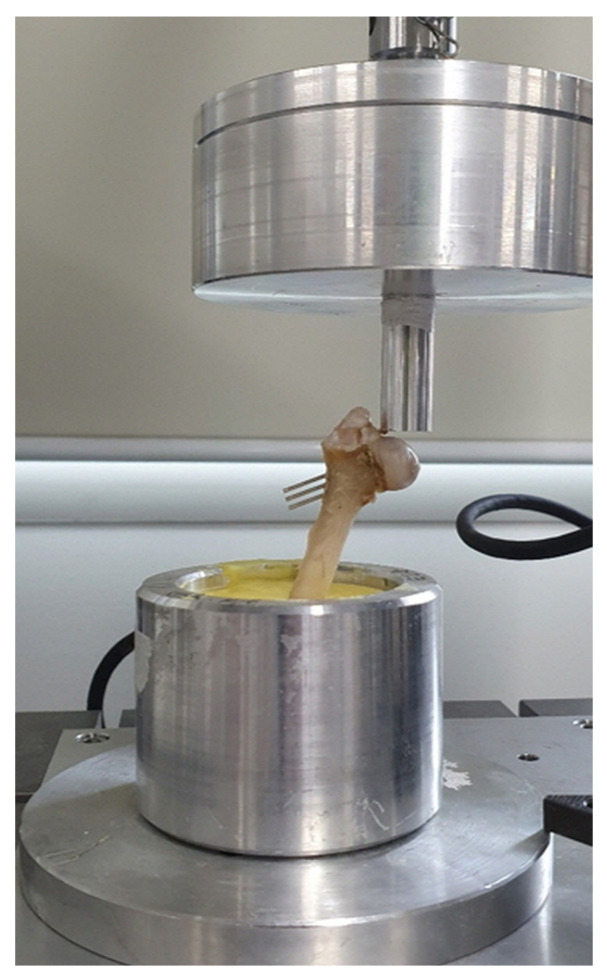
A photograph of static axial loading test. An implanted femur was potted to a cylinder container using a resin and abducted by 20° with respect to the metallic rod mounted on the load cell.

**Table 1 vetsci-10-00285-t001:** Objective measurements of mechanical test to compression force on each fixation.

Group	Yield Point (N)	Stiffness (N/mm)	Displacement (mm)
T	221.26 ± 108.76	54.57 ± 35.34	4.59 ± 2.17
V	126.10 ± 46.30	37.74 ± 16.66	3.85 ± 1.80
*p* values	0.023 *	0.165	0.380

* Statistically significant (*p* < 0.05).

**Table 2 vetsci-10-00285-t002:** Spread and surface area of postoperative K-wires.

Groups	Anteroposterior Spread (%)	Lateral Spread (%)	Surface Area (mm²)
T	43.13 ± 5.82	51.25 ± 4.53	3.62 ± 0.74
V	49.38 ± 11.27	24.25 ± 4.77	0.98 ± 0.27
*p* values	0.113	<0.001 *	<0.001 *

* Statistically significant (*p* < 0.05).

**Table 3 vetsci-10-00285-t003:** Distance from K-wire to femoral neck cortex.

Groups	Distance to Cortex (mm)			Cortical Support (n)
T	PCa	PCr	D	2.75 ± 0.46
	0.56 ± 0.17	0.82 ± 0.18	0.82 ± 0.30	
V	P	M	D	1.75 ± 0.46
	0.68 ±0.20	1.21 ± 0.25	0.72 ± 0.27	
*p* values	n/a			0.007 *

Abbreviations: PCa, proximo-caudal; PCr, proximo-cranial; P, proximal; M, middle; D, distal; n/a, not applicable. * Statistically significant (*p* < 0.05).

## Data Availability

Not applicable.

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
