# Peer review of "Biomechanical Comparison between Inverted Triangle and Vertical Configurations of Three Kirschner Wires for Femoral Neck Fracture Fixation in Dogs: A Cadaveric Study"

_vetsci, 2023, doi:10.3390/vetsci10040285_

Round 1

Reviewer 1 Report

Well designed study, nicely written paper. Good discussion of limitations

Specific

Ln 167 – it would be more likely for the bone to break and the K-wires to bend, than for K-wires to break. Add other reasons for considering a specimen failed

Ln 225 – change point to load

Ln 281 – same comment

Reviewer 2 Report

Dear Authors, I would like to thank you for submitting this interesting paper. You enlightened a peculiar aspect of the Biomechanical configurations of three Kirschner wires for femoral neck fracture fixation in dogs, It was well-described and carefully explained.

Best regards

Row 39. Modify desease with deseases. "Often caused by trauma, fractures of the proximal end of the femur including basilar neck and capital physis are common orthopaedic disease....."
